# Development of an Oral Isoliquiritigenin Self-Nano-Emulsifying Drug Delivery System (ILQ-SNEDDS) for Effective Treatment of Eosinophilic Esophagitis Induced by Food Allergy

**DOI:** 10.3390/ph15121587

**Published:** 2022-12-19

**Authors:** Mingzhuo Cao, Yuan Wang, Heyun Jing, Zeqian Wang, Yijia Meng, Yu Geng, Mingsan Miao, Xiu-Min Li

**Affiliations:** 1Academy of Chinese Medical Sciences, Henan University of Chinese Medicine, Zhengzhou 450058, China; 2College of Pharmacy, Henan University of Chinese Medicine, Zhengzhou 450058, China; 3Department of Pathology, Microbiology and Immunology, and Department of Otolaryngology, New York Medical College, Valhalla, NY 10595, USA

**Keywords:** eosinophilic esophagitis (EoE), isoliquiritigenin (ILQ), self-nano-emulsifying drug delivery system (SNEDDS), increased bioavailability

## Abstract

Isoliquiritigenin (ILQ) is a natural flavonoid with various pharmacological activities. In this study, we optimized the preparation method of self-nano-emulsion-loaded ILQ to further improve its bioavailability based on our previous study. In addition, its effect on the treatment of eosinophilic esophagitis was also evaluated. Combined surfactants and co-surfactants were screened, and the optimal formulation of ILQ-SNEDDS was determined according to droplet size, droplet dispersity index (DDI), and drug loading. The formulation was composed of ethyl oleate (oil phase), Tween 80 & Cremophor EL (surfactant, 7:3), and PEG 400 & 1,2-propylene glycol (cosurfactant, 1:1), with a mass ratio of 3:6:1. Its physicochemical properties, including drug loading, droplets’ size, Zeta potential, appearance, and Fourier transform infrared (FTIR) spectroscopy, were characterized. In vitro release profile, in situ intestinal absorption, and in vivo pharmacokinetics were applied to confirm the improvement of oral ILQ bioavailability by NEDDS. Finally, the efficacy of ILQ-SNEDDS in the treatment of food allergy-induced eosinophilic esophagitis (EOE) was further evaluated. When the ILQ drug loading was 77.9 mg/g, ILQ-SNEDDS could self-assemble into sub-spherical uniform droplets with an average size of about 33.4 ± 2.46 nm (PDI about 0.10 ± 0.05) and a Zeta potential of approximately −10.05 ± 3.23 mV. In situ intestinal absorption showed that optimized SNEDDS significantly increased the apparent permeability coefficient of ILQ by 1.69 times, and the pharmacokinetic parameters also confirmed that SNEDDS sharply increased the max plasma concentration and bioavailability of ILQ by 3.47 and 2.02 times, respectively. ILQ-SNEDDS also significantly improved the apparent signs, allergic index, hypothermia and body weight of EoE model mice. ILQ-SNEDDS treatment significantly reduced the levels of inflammatory cytokines, such as TNF-α, IL-4, and IL-5, and the level of PPE-s-IgE in serum, and significantly inhibited the expression of TGF-β1 in esophageal tissue. SNEDDS significantly improved the solubility and bioavailability of ILQ. Additionally, ILQ-SNEDDS treatment attenuated symptomatology of EoE model mice, which was associated with inhibiting the production of T_H_2 inflammatory cytokines and PPE-s-IgE and the expression of TGF-β1. The above results shows that ILQ-SNEDDS has great potential as a good candidate for the treatment of eosinophilic esophagitis.

## 1. Introduction

Eosinophilic esophagitis (EoE) is a chronic esophageal inflammatory disease associated strongly with food allergy, characterized by significant esophageal eosinophils (Eos) and esophageal dysfunction, such as dysphagia and food impaction [1,2]. EoE imposes, on patients and their families, substantial negative impacts by causing emotional distress (fear and anxiety) and limiting social activities [3]. It has been documented that EoE has been rapidly increasing both in incidence and in prevalence over the last two decades, especially in Western countries [4]. EoE is currently the leading cause of dysphagia and bolus impaction and the second most common cause of chronic esophagitis after gastroesophageal reflux disease [2]. Despite increasing worldwide prevalence, EoE lacks sufficiently effective therapies. Due to the lack of sufficient understanding of the disease, there are no complete epidemiological data on EoE in China so far, but clinical reports of EoE case are gradually increasing significantly. Most EoE cases are in children (mainly 5–10 years old) and adults (mainly 30–50 years old) and clinical symptoms vary by age, with eosinophilic inflammation being seen in children and esophageal remodeling and fibrosis in adults [5,6]. This difference also supports the progression of EoE disease [2,7].

To date, the pharmacotherapy of EoE has highly relied on inhaled and swallowed corticosteroids. Despite their efficacy in suppressing EoE-related inflammation, concerns remain regarding long-term steroid use, such as esophageal candidiasis, abnormal bone mineral density, glaucoma, hyperglycemia, and other serious side effects [8]. It is urgent to find new, highly safe, and effective anti-inflammatory and anti-fibrosis agents for EoE.

Given the important roles of ongoing inflammation and fibrosis in the development of EoE disease, natural small-molecule compounds, such as polyphenols and flavonoids, with anti-inflammatory, antioxidant, and immunomodulatory effects have attracted increasing attention [9,10]. Isoliquiritigenin (ILQ, Figure 1) is a natural flavonoid compound with a chalcone structure, which has strong antioxidant, anti-inflammatory, anti-tumor, and anti-allergic effects [11,12,13,14,15]. In our previous study, we found that ILQ is one of the most potent flavonoids isolated from *Glycyrrhiza* to significantly suppress the Th2 type immune response and the production of eotaxin-1 [14,16,17], both of which greatly contribute to eosinophil inflammation in EoE. However, its poor solubility, fast elimination, and poor in vivo absorption hindered its further application in vivo [18]. Nanotechnology-based drug delivery systems have done good work in increasing the solubility and bioavailability of insoluble drugs in vivo, which has gradually become a consensus. Self-nano-emulsifying drug delivery systems (SNEDDS) have been paid extensive attention for their excellent properties for oral administration, such as spontaneous formation in the gastrointestinal tract, ease of manufacture, and low cost [19].

In our previous study, an oral self-nano-emulsified drug delivery system (SNEDDS)-loaded ILQ was designed to improve its solubility and in vivo bioavailability. However, the use of a high proportion of Tween 80 may result in an increased risk of hemolysis [20]. Currently, compatible surfactants and co-surfactants were used to optimize formulations for further improvement the safety and bioavailability of ILQ-SNEDDS. The appearance, average droplets’ size, zeta potential, and morphology of the optimal ILQ-SNEDDS were characterized. Subsequently, the effects of different drug loadings on the in vitro release and in situ intestinal absorption of ILQ-SNEDDS were also investigated. Finally, the anti-inflammatory and anti-fibrosis effects of ILQ-SNEDDS were evaluated in a mouse model of eosinophilic esophagitis (EoE) induced by food allergy.

## 2. Results and Discussion

### 2.1. The Preparation and Characterization of ILQ-SNEDDS

In our previous study, by screening various excipients with high solubility of ILQ, oil phase, Tween 80, and PEG 400 were selected as oil phase, surfactant, and co-surfactant, respectively, with a weight ratio of 3:6:1. The high doses of Tween 80 in this formulation increase the risk of hemolysis toxicity of ILQ-SNEDDS administered in vivo. Therefore, in order to improve the safety of the nano-emulsion, this study intended to further improve the solubility of ILQ and decrease the dosage of Tween 80 without reducing the bioavailability of ILQ by screening other suitable mixed surfactants and co-surfactants.

Since the solubility of ILQ was nearly the same in Tween 80 and Cremophor EL, Cremophor EL was preferred as the mixed surfactant. The appropriate proportions between Tween 80 and Cremophor EL were further determined by a single factor factorial experiment. Maintaining the weight ratio of oil phase (ethyl oleate): mixed surfactant and co-surfactant (PEG-400) at 3:6:1, the ratios of Tween 80 and Cremophor EL were set at 9:1, 8:2, 7:3, 6:4, and 5:5. each ILQ-SNEDDS was prepared according to our previous method. The optimum ratio of Tween 80 to Cremophor EL was determined by comparing ILQ solubility, droplets’ size, polydispersity index, and ξ potential. As shown in Figure 2A, each ILQ-SNEDDS showed both increased ILQ loading and droplets’ size after the addition of Cremophor EL. When the ratio was 9:1, the drug loading of ILQ and its droplets’ size were both the largest. As the Cremophor EL ratio continued to increase, the drug loading of ILQ and droplets’ size both decreased a bit. The ratios of 8:2 and 7:3 showed high ILQ loading and small droplet size and polydispersity index, and was selected for the next experiment.

Among the commonly used surfactants, the solubility of ILQ in 1,3-propanediol is the highest. However, droplet sizes of the nano-emulsion formed only with 1,2-propanediol as the cosurfactant were all more than 100 nm. Therefore, 1,2-Propanediol was considered to be used in combination with PEG 400 as a cosurfactant. A single factor experiment was used to further screen out the best ratio of the two. Ethyl oleate (oil phase):Tween80 and Cremophor EL (surfactant):1,2-propanediol and PEG 400 (cosurfactant) mass ratio was still set to 3:6:1, in which the ratios of Tween 80 to Cremophor EL (as co-surfactants) were set at 8:2, 7:3, and 5:5. The mass ratios of PEG 400 to 1,2-propanediol were set at 1:0, 1:1, and 1:2. The ILQ drug loading and droplet sizes of the prepared nano-emulsions are shown in Figure 2B. The results showed that the drug loading of ILQ increased when a certain proportion of 1,2-propanediol was added. But when the mass ratio of 1,2-propanediol to PEG 400 was set at 2:1, the droplet size of SNEDDS increased significantly to above 100 nm. Therefore, the mass ratio of 1,2-propanediol to PEG 400 (mixed cosurfactant) was determined to be 1:1.

Based on our previous study, a three-factor (Factor A, percentage of oil phase; Factor B, mass ratio of Tween80: Cremophor EL (mixed Surfactant); Factor C, the Km ratio of surfactant to co-surfactant) and three-level orthogonal experimental design were used to screen out the best formulation by comparing the drug loading of ILQ, droplets’ size (polydispersity index and ξ potential) of the formed ILQ-SNEDDS. The results are shown in Table 1. The results of ILQ loading were analyzed first. The Km value of the ratio of surfactant and co-surfactant had the greatest influence on drug loading (R value was 19.91), followed by the percentage of oil phase (R value was 11.69). Mass ratio of Tween80 to Cremophor EL had the least effect on the results (R value of 4.83). Then, considering the droplets’ size factor, when the Km value was 5:2, the droplets’ sizes of the prepared nano-emulsion were not stable, which might be quite small (12 nm) or large (190 nm, 230 nm). Therefore, the finalized optimal SNEDDS formulation was oil phase (ethyl oleate), mixed surfactant (Tween 80: Cremophor EL, 7:3), and mixed cosurfactant (PEG 400:1, 2-propanediol = 1:1), with a mass ratio of 3:6:1. After 10-fold dilution with ultrapure water, they could quickly form a clear and transparent nano-emulsion in less than 2 min (Figure 3A–C). An interesting phenomenon was found during the study that the droplet size only increased slightly when the drug loading was lower than 80 mg/g under the optimized formulation conditions. The drug loading of ILQ-SNEDDS was 77.9 mg/g, and the encapsulation efficiency was 92.50% ± 0.45. As shown in Figure 3, the average droplet size was 33.40 ± 2.46 nm with a PDI of 0.10 ± 0.05 and its Zeta potential was −10.05 ± 3.23 mV. The results of TEM images of ILQ-SNEDDS (Figure 3D) were highly consistent with the DLS results. Most droplets were spherical in shape and uniform in size, ranging from 28–38 nm, with an average droplet size of 32.3 nm (Figure 3E). However, when the drug loading further increased, the droplet size was increased significantly and was positively correlated with drug loading. When the drug loading increased to 90 mg/g, the droplet size had increased to 140 nm with a broadened PDI, and when it further increased to 132 mg/g, the droplet size had increased to more than 240 nm.

On the FTIR spectrum (Figure 3F), the characteristic absorption peaks at 3426 cm^−1^ (O-H stretching vibration, disappeared), 2923 and 2858 cm^−1^ (νC-H stretching vibration, decreased), 1630 cm^−1^ (νC=C, increased), 1514 cm^−1^ (νC-H, increased), and 1110 cm^−1^ (νC-O stretching vibration, decreased) showed remarkable changes, indicating the increased interaction force between ILQ and SNEDDS.

### 2.2. In Vitro Release Profiles and Stability of ILQ-SNEDDS

In our previous study, we found that ILQ-SNEDDS showed robustness to dilution and good stability for storage. Therefore, we focused on the release behavior of ILQ-SNEDDS with different drug loading in the current study.

Figure 4A showed the release profiles of ILQ from a nano-emulsion with different drug loading. The results demonstrated that there was no significant difference in the cumulative release percentage for 12 h between ILQ-SNEDDS groups with different drug loading. More than 75% of ILQ were released from ILQ-SNEDD with a droplet size less than 33 nm. In addition, they were 20% higher than that of the ILQ suspension. However, ILQ-delayed release in simulated gastric juice was only observed in ILQ-SNEDDS groups with a droplet size of less than 33 nm and was negatively correlated with droplet sizes in comparison to the free ILQ-suspension. ILQ-SNEDDS with the largest droplet size (~240 nm) showed similar release characteristics with ILQ suspension in 1% Tween 80. Figure 4B showed the appearance of a nano-emulsion diluted 10 times and 100 times under different drug loading. These results suggested that ILQ-SNEDDS with small droplet size could successfully incorporate ILQ in bilayer emulsion droplets, thus contributing to much more ILQ release than the free ILQ suspension [21].

SNEDDS is a thermodynamically stable system. The optimized ILQ-SNEDDS could withstand dilution with a large amount of simulated gastric juice (pH1.2 hydrochloric acid) and intestinal juice (pH 6.8 phosphate buffer), and its droplet size, PDI, and zeta potential showed minor changes under different harsh dilution conditions, indicating the good dilution stability of optimized ILQ-SNEDDS (Table 2). The optimized ILQ-SNEDDS was stored at 4 °C or 37 °C for more than one month, and its appearance did not show any change. However, with the extension of storage time to 2 months, it might coalesce, showing enlarged droplet sizes and reduced Zeta potentials (Table 2), especially under 37 °C.

### 2.3. Intestinal Absorption In Situ of ILQ-SNEDDS

In situ perfusion was used to evaluate the gastrointestinal absorption improvement potential of SNEDDS-ILQ preparations with different drug loading. As shown in Figure 4C, SNEDDS significantly increased the absorption of ILQ in the jejunum as expected, compared with free ILQ. However, ILQ drug loading could also significantly affect the absorption rate. The absorption rate of ILQ-SNEDDS with 75 mg/g was the highest and comparable to the drug loading of 35 mg/g, reaching 23.45%, which was 9.02% higher than that of free ILQ (*p* < 0.01), while ILQ-SNEDDS with 95 mg/g drug loading only increased by 3.03%. Parameters representing absorption enhancement, such as absorption rate constant (Ka), apparent permeability coefficient (Papp), and enhancement ratio (ER), were calculated and listed in Table 3. The results of in situ perfusion were consistent with the results of in vitro release, indicating that under the conditions of ensuring the small droplet size, properly increasing the drug loading would not affect the role of SNEDDS in promoting the absorption of ILQ. However, the excessive increase in drug loading would lead to a significant decrease in the absorption rate with the increase in droplet size.

In a previous study, it was reported that a surfactant with an HLB value ranging from 10 to 17 in SNEDDS formulation plays a crucial role in enhancing drug absorption and intestinal penetration, which may be associated with the fact that surfactants could alter the mucus structure and enhance the hydrophilicity of the intestinal microenvironment [22,23]. In the current study, the co-surfactant of Tween 80 (with HLB 15) and Cremophor EL (with HLB 13.5) in the SNEDDS undoubtedly increased the absorption rate constant and apparent permeability coefficient of ILQ compared to free ILQ suspension. However, the droplet size was another important reason for the enhancement of ILQ permeation since the smaller droplet can easily permeate via the intestinal wall. If SNEDDS is overloaded with ILQ, its looser surfactant coating resulted in a larger droplet size and significantly lower permeability coefficient. Therefore, in the following experiments, the ILQ-SNEDDS with a drug loading of 75 mg/g was selected for in vivo experimental study.

### 2.4. Pharmacokinetic Study of ILQ-SNEDDS

Rats were given a single dose of 35 mg/kg free ILQ (dissolved in 1% Tween 80) and ILQ-SNEDDS (28.9 nm, equivalent to free ILQ). The plasma drug concentration–time profiles are presented in Figure 5, and the pharmacokinetic parameters were calculated by using a non-ventricular analysis and listed in Table 4. As shown in Figure 5, the maximum plasma concentration (C_max_) of ILQ in the nano-emulsion group was 1.52 μg/mL, 3.47 times higher than that in the free ILQ group. Moreover, usage of SNEDDS led to a significant reduction in T_max_ (0.5 h vs. 0.75 h) and a statistically significant increase in AUC_0–t_ compared to the free ILQ suspension.

Due to its poor dissolution capacity and low intestinal permeability, the free ILQ suspension showed poor bioavailability, while SNEDDS could improve the oral bioavailability of ILQ. The AUC_0–24_ of the ILQ-SNEDDS was 3.80 μg·h·mL^−1^, 2.02 times higher than that of free ILQ suspension, which was correlated with the result of the in vitro release study and in situ perfusion. In summary, the increased bioavailability of ILQ-SNEDDS was strongly associated with the following two factors: (ⅰ) SNEDDS could significantly increase the solubility of ILQ and ensure its dissolution, thereby improving the dissolution limiting step of ILQ [24,25]; (ii) the nano-emulsion with small droplet size could promote the intestinal lymphatic transport of ILQ [26,27]. Therefore, the C_max_ and AUC_0–24_ of ILQ-SNEDDS increased by 3.47 times and 2.02 times, respectively, compared to the free ILQ suspension.

### 2.5. Hemolytic Toxicity of ILQ-SNEDDS

Tween-80 is widely used as a solubilizer and stabilizer in injection and oral preparations with a commonly used dosage of about 1–2%, including docetaxel and some vaccines. However, the hemolysis problem and other toxicities, including sensory neuropathy, nephrotoxicity, and hypersensitivity reactions caused by excessive use of Tween-80, have also attracted extensive attention, and the safe dosage of Tween-80 has become the focus of its use [20,28].

In this study, Tween-80 was still used as the major surfactant, but its dosage was reduced by more than 30% in the optimized formula. As shown in Figure 6, when the dose of ILQ-SNEDDS was under 50 μg/mL, no hemolysis could be observed. Within the effective therapeutic dose of ILQ-SNEDDS, the highest plasma concentration of ILQ was only about 1.65 μg/mL, the hemolysis toxicity caused by Tween 80 was unnoticeable. The single oral dose was even increased to 100 mg/kg and no obvious toxicity was found in rats. Since viable alternatives to Tween 80 remain unsolved, the toxicity caused by excessive use of Tween-80 should still be paid sufficient attention. The standardized use of Tween-80 is an essential method for the safe use of nano-emulsion.

### 2.6. Pharmacodynamics Effects and Underlying Mechanism of ILQ-SNEDDS on EOE-Like Model Mice

#### 2.6.1. ILQ-SNEDDS Alleviated Weight Loss and Hypothermia in EOE-Like Model Mice

A PPE-induced food allergy-like EOE mouse model was constructed according to the method (shown in Figure 7A) in a previous study [1] with minor modifications to evaluate the therapeutic effect of ILQ-SNEDDS. EOE mice daily received oral treatment with ILQ-SNEDDS at a 20 mg/kg equivalent ILQ dose for one month. In our previous study, ILQ-SMEDDS was developed to treat asthma, and the ILQ-SMEDDS group at the dose of 10 mg/kg showed a better anti-asthma effect than that of the ILQ suspension group at a dose of 20 mg/kg [29]. We made a first attempt to use ILQ to treat EOE, choosing ILQ-SNEDDS with higher bioavailability instead of free ILQ, which is also a limitation of the current study.

We combined oral PPE challenge way and skin PPE challenge way to enhance the symptoms of experimental eosinophilic esophagitis. The PPE-induced EOE mouse model could well simulate the typical symptoms and pathological changes of EoE with typical characteristics of food allergy. Compared with the mice in the naïve group, all the model mice showed significant weight loss, decreased activity, poor hair luster, loose stool, spleen swelling, and responses of anaphylaxis, such as increased scratching behavior, hypothermia, cyanosis, difficulty breathing, and other severe hypersensitivity responses. In addition to the above allergic symptoms, the model mice also had symptoms of ear scabbing or even thickening. The above symptoms were mainly consistent with the clinicopathological changes of EoE. As shown in Figure 7B, the mean weight of sham-treated mice decreased significantly after each PPE challenge, which was 0.63 g lower than before the experiment after the first PPE challenge. Additionally, it had decreased by 2.71 g by the end of the experiment. Although the weight of the mice treated with ILQ-SNEDDS also decreased after PPE challenges, the weight of mice even increased by 0.63 g after four PPE challenges and only decreased by 0.29 g after the last high-dose PPE challenge. Furthermore, all sham-treated mice developed anaphylaxis with a mean symptom score of 2.71. Treatment with ILQ-SNEDDS could effectively reverse the pathogenesis of esophagitis and greatly alleviated these symptoms. The allergic symptom scores of mice treated with ILQ-SNEDDS were significantly lower than those of the sham mice (*p* < 0.05, Figure 7D), with no significant difference from dexamethasone-treated mice.

H&E staining results directly showed the pathological changes of EOE model mice. As shown in Figure 7C, the esophageal tissue of mice in the sham group was altered in association with the disease; the esophageal wall was significantly thickened, swollen, and congested, the lamina propria was elongated, the upper lamina propria was fibrotic, and the infiltration of inflammatory cells and eosinophils around the esophagus was significantly increased compared to the naïve mice. ILQ-SNEDDS and Dex treatment greatly relieved the above symptoms.

Figure 7E showed the changes in body temperature of all mice within 40 min after the last challenge. Core body temperature of all model mice dropped within 30 min after the challenge and recovered after 40 min. The mean body temperature of sham-treated mice was the lowest at 30 min, which was 33.92 ± 1.01 °C, and 2.08 °C lower than that of the naïve mice (*p* < 0.001). ILQ-SNEDDS and dexamethasone treatment could alleviate hypothermia in model mice, and their mean body temperature was 35.18 ± 0.54 °C and 34.84 ± 0.73 °C (*p* < 0.05, and *p* = 0.15, compared with the sham group). Anaphylaxis, such as food allergy and EoE, is a systemic disease triggered by the release of multiple inflammatory cells and might cause a variety of viscera damage [1]. The results of pathological section also showed that the intestinal structure of the sham mice was abnormal, damaged, and accompanied by a large number of inflammatory cells’ infiltration. In addition, the sham mice also showed significant pulmonary inflammation and splenomegaly, with a significant increase in lung index (151.5%, *p* < 0.001, Figure 7F) and spleen index (208.0%, *p* < 0.001, Figure 7G) compared with the naïve mice. After treatment with ILQ-SNEDDS, the lung index and spleen index of mice decreased by 15.3% and 12.0%, respectively (*p* < 0.05, *p* = 0.1), indicating that ILQ-SNEDDS treatment could improve inflammation in the small intestine, spleen, and lung of mice to a certain extent.

#### 2.6.2. ILQ-SNEDDS Reduced the Levels of IL-4, IL-5 and TNF-α in Peripheral Blood of EoE-Like Model Mice

Increasing evidence shows that EoE is the late manifestation of anaphylaxis progression, which usually starts from atopic dermatitis and develops into IgE-mediated food allergy, asthma, allergic rhinitis, and EoE [30]. Moreover, as a progressive disease, if not treated, the persistent inflammation generated during the active EoE will accelerate the progress of esophageal stenosis and fibrosis remodeling, leading to irreversible esophageal function damage in patients [31]. In the complex pathogenesis of EoE, multiple immune cells, such as T_H_2 cells, eosinophils, and mast cells, are involved and jointly contribute to the inflammation produced in the active EoE [31,32].

As shown in Figure 8A–C, the levels of TNF-α, IL-4, and IL-5 in the serum of sham mice were sharply increased compared with those in naïve mice (*p* < 0.0001, *p* < 0.0001, and *p* < 0.0001). TNF-α is one of the earliest and most important inflammatory mediators in the process of inflammatory response [33]. TNF-α levels in the serum in the sham group reached 484.57 ng/mL. ILQ-SNEDDS and dexamethasone treatment could significantly reduce the above inflammatory cytokines compared with the sham group. The levels of TNF-α, IL-4, and IL-5 were decreased by 33.3% (*p* < 0.05), 56.7% (*p* < 0.001), and 39.6% (*p* < 0.01), respectively, after ILQ-SNEDDS treatment (Figure 8B,C).

#### 2.6.3. ILQ-SNEDDS Suppressed the Expression of TGF-β1

In addition to the above inflammatory cytokines produced by immune cells, TGF-β1-mediated esophageal inflammation and fibrosis also played a critical role in the occurrence and development of EoE [5,34,35]. One study found that blocking TGF-β1 and MAPK signaling could inhibit the secretion of fibronectin and type I collagen by esophageal fibroblasts and muscle cells, thus relieving esophageal fibrosis [36]. As shown in Figure 8D, the TGF-β1 positive expression was yellowish brown and diffusely distributed in the cytoplasm of esophageal tissue. The TGF-β1 positive expression level in the sham group was much higher than that of the naïve group. After ILQ-SNEDDS treatment, the TGF-β1 positive expression level was obviously decreased.

#### 2.6.4. ILQ-SNEDDS Reduced PPE-s-IgE Production in Peripheral Blood of EoE-Like Model Mice

Whether specific IgE plays an active role in the pathogenesis of disease or whether it is only a prognostic marker remains unclear. However, intensive studies have also confirmed that IgE is indeed closely associated with EoE [32].

Figure 8E showed the results of PPE-s-IgE levels measured by ELISA. Compared with the naïve group, the serum PPE-s-IgE in the sham group was significantly increased (*p* < 0.0001), with an average concentration of 369.89 ng/mL. ILQ-SNEDDS treatment could depress serum PPE-s-IgE production, which was significantly decreased by about 40% compared with that in the sham group (*p* < 0.01).

These physiological conditions and symptoms associated with EoE disease were gradually improved following the ILQ-SNEDDS treatment. However, although the safety of natural small molecule ILQ is significantly higher than that of dexamethasone, its improvement effect on EOE clinical symptoms is still inferior to that of dexamethasone. In view of the good anti-inflammatory and antioxidant effects of ILQ, it has significant potential in the treatment of asthma, food allergy, and EoE. We will continue to look for other phytonanotechnologies [37] that can improve the bioavailability of ILQ more efficiently in our subsequent research.

This study has several limitations. EoE is commonly associated with concomitant atopic diseases including atopic dermatitis, food allergy, and asthma. There were no significant differences between sexes in mid/proximal or distal esophageal eosinophil count in both adults and children [38]. Although EOE worldwide is predominant in male children and adults [38], we used female mice in our study according to the literature [1], because female mice experience more airway remodeling and T_H_2 inflammation compared with male mice [39,40]. However, due to the differences in hormone levels and metabolism between men and women, to mice gender should be properly considered in EOE pathogenesis and drug absorption [41].

## 3. Materials and Methods

Isoliquiritigenin (ILQ, 98.0%) was purchased from Herb Purify Co., Ltd. (Chengdu, China). Ethyl oleate (EO), 1,3-propanediol, acetanilide Cremophor EL, urethane, and phenol red were purchased from Aladdin (Shanghai, China). Tween 80 and Al (OH)_3_ gels were obtained from Sigma Aldrich (Shanghai, China). PEG 400 was obtained from Yipusheng Pharmaceutical Co., Ltd. (Ji’an, China).

### 3.1. Animals

Sprague Dawley (SD) rats (200 ± 20 g) and BALB/c mice (6–8 weeks old) were obtained from Shandong Laboratory Animal Center (Peng-Yue Laboratory Animals, Jinan, China). All mice were housed at 22 ± 2 °C in cages within laminar airflow hoods in a specific pathogen-free room with a 12-h light/12-h dark cycle and fed autoclaved chow and water ad libitum.

### 3.2. Preparation and Characterization of ILQ-SNEDDS

#### 3.2.1. HPLC Method

The HPLC (e2695 system, Waters, Eschborn, Germany) method was established for the analysis of ILQ as previously reported with some modifications [16]. The detailed conditions were as follows: acetonitrile–water mixture (40/60, *v*/*v*) was used as mobile phase, with HPLC temperature set at 30 °C, wavelength at 372 nm, and flow rate 1.0 mL/min. The solubility of ILQ in surfactants was measured using the method reported in our previous study [29].

#### 3.2.2. Optimization, Preparation, and Characterization of the ILQ-SNEDDS

In this study, we aimed to decrease the dosage of Tween 80. The use of mixed cosurfactants could directly reduce the dosage of Tween 80 on the one hand, and on the other hand, it could further indirectly reduce the dosage of Tween 80 by increasing the solubility of ILQ.

ILQ-SNEDDS was prepared using the method reported in our previous study. All emulsion components were mixed in pre-designed proportions and sonicated for 10 min to produce a homogeneous liquid. Then, excess ILQ was added to the above liquid and sonicated for 10 min. After that, the mixture was oscillated overnight at 100 rpm and 37 °C in an oscillation (Anjing Equipment, Shanghai, China). Finally, the undissolved ILQ precipitate was removed by centrifugation (Microfuge 20R, Beckman Coulter, Pasadena, CA, USA) at 8000 rpm for 30 min. Additionally, the transparent ILQ-SNEDDS was stirred gently with water (100-fold, *v*/*v*) drop by drop. The size, polydispersity index, and Zeta potential of ILQ-SNEDDS droplets were determined using a NanoBrook90PlusPALS meter (Brookhaven Instruments Corporation, Holtsville, NY, USA) at 25 °C.

Single-factor and orthogonal experimental designs were used to screen suitable ratios of mixed surfactants, mixed co-cosurfactants, and best prescription. Firstly, the weight ratio of ethyl oleate (EO, oil), Tween 80 (surfactant), and PEG-400 (co-surfactant) was fixed as 3:6:1, Tween 80 was replaced with different ratios of Tween 80 and Cremophor EL, and the droplets’ size and drug loading were obtained using the above methods. Then, the optimal ratios of Tween 80 and Cremophor EL in the previous step were chosen for further screening of the optimum ratio of PEG-400 and 1,3-propanediol. Finally, a three-factor and three-level orthogonal experiment was performed to explore the best formulation.

The size and morphology of optimized ILQ-SNEDDS were obtained by transmission scanning electron microscope (TEM-1400, Tokyo, Japan) after staining with 2% phosphato-tungstic acid (PTA). Drug encapsulation efficiency (EE) and drug loading (DL) were measured by the centrifugation method previously reported in the literature with some modification [42]. The new prepared ILQ-SNEDDS was diluted with methanol. The amount of ILQ was determined by HPLC after being filtered through a 0.22 μm filter membrane. The encapsulation efficiency and drug loading (DL) were calculated as follows:(1)$$EE\left(\%\right)=\frac{W_{\mathrm{Loaded}}}{W_{\mathrm{Total}-\mathrm{added}}\;}\times 100\%$$
(2)$$DL=\frac{W_{\mathrm{Loaded}}}{W_{\mathrm{Emulsion}}}$$

In which *W*_loaded_ and *W*_total-added_ are the weight of encapsulated ILQ, the total amount of ILQ added, and the weight of emulsion added, respectively.

The IR spectra of ILQ-SNEDDS, ILQ, and mixture of ILQ and SNEDDS were acquired on the Fourier-transform infrared spectrometer (FT-IR) (Spectrum 100, Platinum Elmer, Waltham, MA, USA) by using a potassium bromide pressed disk method.

#### 3.2.3. Stability of ILQ-SNEDDS

The stability of ILQ-SNEDDS under certain storage times and pH values was investigated, and the changes in emulsion appearance and droplets’ size were recorded. ILQ-SNEDDS samples after dilution with 100 times ultra-pure water were stored in vials at 37 °C or 4 °C for 2 months. At a set time of 0, 0.5, 1, and 2 months, the droplets’ size, PDIs, and zeta potentials were measured.

Tolerating the physiological pH of the gastrointestinal tract is very important for oral nano-emulsions to improve the bioavailability of drugs. The newly prepared ILQ-SNEDDS were diluted with simulated gastric fluid (pH 1.2 HCL) and intestinal fluid (pH 6.8 PBS) within a predetermined range, and the droplets’ size, PDIs, and ζ potential were measured.

#### 3.2.4. In Vitro Release Profile

The in vitro release of ILQ in ILQ-SNEDDS with different droplets’ size in the simulated gastric and intestinal fluid was performed using the dialysis-diffusion method according to the Chinese Pharmacopoeia (2020) with slight modification. ILQ suspension (5 mg ILQ in 2 mL of 2.0% Tween 80 liquid) and four ILQ-SNEDDSs (2 mL nano-emulsions with equivalent amount of ILQ) were placed in dialysis bags (weight cut-off 5000 Da). They were then placed in 250 mL release medium containing 1% Tween 80 (pH 1.2 HCl for the first 2 h and pH 6.8 PBS for the last 10 h) at 37 °C and oscillated at 150 rpm. Dialysate (2 mL each) was taken at scheduled time points and replenished quickly with fresh medium of equal volume. The release amount of ILQ in dialysate was determined by HPLC.

### 3.3. Absorption In Situ and Pharmacokinetic In Vivo Studies

#### Intestinal Absorption In Situ

The effects of different droplet sizes of ILQ-SNEDDS on the percent absorption (AP), absorption rate constant (Ka), and apparent permeability coefficient (Papp) of ILQ intestinal absorption were investigated using a rat in situ perfusion model according to the reported method with minor modifications [43,44]. Briefly, anesthetized rats were restrained in the supine position and maintained at normothermia using an infrared light. Upon confirmation of the loss of pain reflex, its abdomen was incised with an approximately 3 cm midline longitudinal incision. The distal jejunum was intubated immediately with two glass tubes (3.5 mm, OD) and ligated with surgical silk suture. Selected intestinal segments were gently rinsed with 5 mL of pre-warmed pH 7.4 PBS to remove fecal residue and debris, and then attached to the perfusion assembly consisting of a 2-head peristaltic pump (L100-1F/DG-2, Baoding Extended Precision Pump Co., Ltd., Baoding, China). One hundred milliliters of ILQ-SNEDDS with different droplets’ size or ILQ suspension with an equivalent initial ILQ concentration were perfused through the selected intestinal segment at a flow rate of 6 mL/min for 10 min. The flow rate was subsequently adjusted to 3 mL/min, and after equilibration for an additional 15 min, the solution volume was recorded as the 0 min volume. Each perfusion experiment lasted for 1.75 h, and samples were taken at a preset interval of 15 min. At the end of the experiment, the radius and lengths of the perfused intestinal segment and the amount of ILQ remaining in each sample were measured. The absorption rate (AP), absorption rate constant (Ka, min^−1^), apparent permeability coefficient (Papp, cm^2^·min^−1^), and enhancement ratio (ER) of ILQ-SNEDDS with different droplets size or ILQ-Suspension were calculated as follows:(3)$$\mathrm{AP}=\frac{C_{0}V_{0}-C_{0}V}{C_{0}V_{0}}\times 100\%$$
(4)$${\ln\;X}_{t}{=\ln X}_{0}{-\;K}_{a}\times \;t$$
(5)$$P_{\mathrm{app}}{=K}_{a}/A$$
(6)$$\mathrm{ER}=\frac{P_{\mathrm{app}}\;\mathrm{of}\;\mathrm{ILQ}-\mathrm{SNEDDS}}{P_{\mathrm{app}}\;\mathrm{of}\;\mathrm{free}\;\mathrm{ILQ}\;\mathrm{Suspension}}\times 100\%$$

In which *C*_0_ (or *C_t_*) and *V*_0_ (or *V_t_*) were the concentration and volume of ILQ-SNEDDS (or free ILQ-suspension) in perfusate at 0 (or t) h, respectively. *X_t_* and *X*_0_ indicated the amounts of ILQ remaining in the perfusate at t h and 0 h, respectively. A (cm^2^) was the surface area of the perfused intestinal segment (Length × diameter of intestinal).

### 3.4. Pharmacokinetic Studies

After one week of domestication, 20 female SD rats were randomly divided into two groups. All the animals were fasted overnight before the experiment with free access to water.

Rats were orally administered either free ILQ suspension (in PBS containing 1% Tween 80) or ILQ-SNEDDS (ILQ equivalent) at 35 mg/kg. We collected 0.5 mL of blood from the retro-orbital plexus at each set time point (0, 0.25, 0.5, 0.75, 1, 1.5, 2, 4, 8, 12, 24 h) after administration. Plasma was obtained by centrifugation at 3000 rpm for 15 min and stored at −80 °C for further HPLC analysis. Then, 100 μL of plasma sample was mixed with 15 μL of acetanilide (50 μg/mL, internal standard) and deproteinized by adding 600 μL of acetonitrile for 15 min. After centrifugation at 12,000 rpm for 25 min, the supernatant was dried at room temperature in a concentrator (Eppendorf, Hamburg, Germany). The samples were dissolved in mobile phase and analyzed by HPLC.

Non-compartmental analysis was performed using PK Solver 2.0 software (a plug-in program for pharmacokinetic analysis in Microsoft Excel) to obtain key pharmacokinetic parameters, including Lambda_z (h^−1^, terminal elimination rate constant), T_max_ (h, time to reach maximum plasma concentration), C_max_ (μg·mL^−1^, maximum plasma concentration), AUC_0–24_ (μg·mL^−1^, area under the plasma concentration time curve), Vz/F (L·kg^−1^, apparent volume of distribution), and T_1/2_ (h, half-life).

### 3.5. Hemolytic Toxicity of ILQ-SNEDDS

We prepared 5 × 10^8^ /mL Red Blood Cells (RBC) suspension according to the method reported in our previous study [29,45]. First, 0.5 mL of 5, 10, 25, 50 μg/mL ILQ-SNEDDS and equivalent of Tween 80 were added to equal volume of RBC suspension, and 1% Triton X100 and PBS were used as positive and negative controls, respectively (*n* = 3). After gently shaking, the RBC suspension was incubated at 37 °C for 3 h. Afterwards, each sample was centrifuged at 3500 rpm/min at room temperature for 5 min. Then, 100 µL of supernatant of each sample was transferred to a 96-well plate and the absorbance at 570 nm was recorded using a microplate reader (SpectraMax iD3, Molecular Devices, San Jose, CA, USA). Percentage of hemolysis was calculated with the following equation:Hemolysis% = (OD_sample_ − OD_PBS_)/(OD_1% Triton-100_ − OD_PBS_) × 100%(7)

### 3.6. Establishment of PPE-Induced EOE Model

The peanut protein extract (PPE)-induced EoE model was established using a previously reported method with minor modification [1,34]. The BALB/c mice were randomly divided into four groups according to the random number table method after adaptive feeding for 1 week, which were designated as naïve group, sham group, dexamethasone control group, and nano-emulsion treatment group, respectively. PPE was prepared by PBS extraction after completely degreasing with acetone, following the method used in our laboratory before.

Except for the naïve group, the BALB/c mice were sensitized by intraperitoneal (i.p.) injections of 0.2 mg PPE adsorbed to 2 mg Alum adjuvant in 200μL PBS on the 0th, 7th, 14th, and 21st days. Meanwhile, they were sensitized by applying 20 mg of calcipotriol ointment containing 1% PPE to the left ear from the days 1–7 and 21–28. Then, they were challenged orally with 5 mg PPE in 100 µL PBS on days 28, 35, 42, and 49, and 10 mg PPE in 100 µL PBS on day 56 (for a total of 5 challenges). All mice were intragastrically treated for 5 weeks. The naïve group was injected, applied, or treated with an equal volume of PBS.

### 3.7. Pharmacodynamic Effect Evaluation

General epigenetic indications’ observation, serum PPE-specific-IgE (PPE-s-IgE), IL-4, IL-5, TNF-α cytokines, and esophagus, small intestine, lung histopathological analyses were used to evaluate the pharmacodynamic effect of ILQ-SNEDDS on food allergy-induced EoE.

Epigenetic indications included body weight, other behaviors during the entire experiment, rectal temperature, and allergy index after the last challenges, as well as spleen and lung index after sacrifice.

Allergy indices were observed over 40 min and grades 0–5 was used to score the reactions by blind method as previously described [46,47] with minor modifications. The criteria were as follows: 0—asymptomatic; 1—head and nose disturbance; 2—diarrhea, edema around eyes and mouth, erect hair, and decreased activity frequency; 3—cyanosis, dyspnea and decreased activity frequency, and drop in temperature; 4—no reaction or only mild reaction, hypothermia, tremor, and spasm; 5—death. Rectal temperatures were measured using a rectal probe at 0, 10, 20, 30, and 40 min.

Blood was collected at the end of the experiment. The levels of PPE-s-IgE, IL-4, IL-5, and TNF-α (pharmingen, BD Biosciences, New York, USA) were determined by ELISA according to the manufacturer’s instructions. The mice were sacrificed, the tissues of esophagus and small intestine were taken and fixed with 4% paraformaldehyde, paraffin-embedded, and sectionalized in 5μm thick sections using a Leica RM2235 microtome (Leica, Nussloch, Germany). The pathological specimens were stained with hematoxylin and eosin (H&E). The inflammatory infiltration, tissue damage, villus deformation, and edema of intestinal epithelial cells in the tissues were observed by an inverted microscope (Nikon, Tokyo, Japan).

TGF-β1 Immunohistochemistry: The paraffin section of the esophagus tissue was dewaxed and fixed, incubated with TGF-β1 antibody (Immunoway, Wuhan, China), and then antibody II, and then the slices were stained and sealed. The positive expression of TGF-β1 was observed with an inverted microscope.

### 3.8. Statistical Analysis

All data were represented as mean ± standard deviation (SD). All inter-group statistical analyses were performed by using unpaired *t* test (Mann–Whitney test, *p* smaller than 0.05 considered statistically significant) by GraphPad Prism 9 software (GraphPad Software, Inc., La Jolla, CA, USA).

## 4. Conclusions

In our previous study, the good properties of ILQ-SNEDDS improved the bioavailability and anti-asthma activity of ILQ. In the current study, and we further optimized the formulation of SNEDDS to reduce the Tween 80 dosage and made a first attempt to use ILQ to treat EOE. The optimized formula of ILQ-SNEDDS consisted of an oil phase (ethyl oleate), surfactant (Tween 80: Cremophor EL = 7:3), and cosurfactant (PEG 400:1, 2-propylene glycol = 1:1), with a mass ratio of 3:6:1, and it showed good biophysical properties, including small droplet size, appropriate negative Zeta potential, high drug content, and good stability. These favorable properties of ILQ-SNEDDS improve the bioavailability of ILQ and made ILQ show excellent anti-EoE activity by inhibiting the production of T_H_2-inflammatory cytokines and PPE-s-IgE, and the expression of TGF-β1 in EoE model mice. This study provides a promising new drug candidate for the treatment of eosinophilic esophagitis.

## Figures and Tables

**Figure 1 pharmaceuticals-15-01587-f001:**
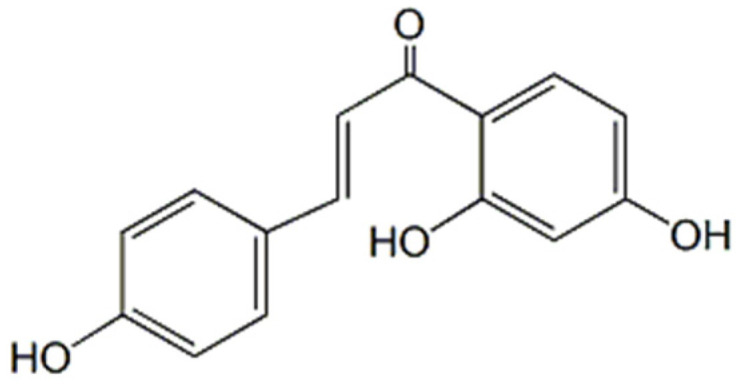
The structure of isoliquiritigenin (ILQ).

**Figure 2 pharmaceuticals-15-01587-f002:**
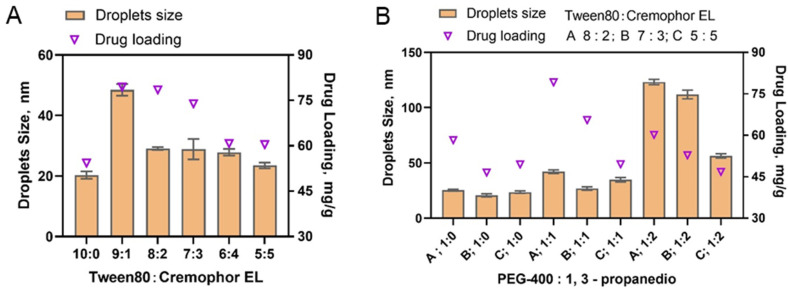
A single factor experiment was used to screen suitable mixed surfactant. (**A**) Screening of mixed surfactants; (**B**) screening of mixed co-surfactants.

**Figure 3 pharmaceuticals-15-01587-f003:**
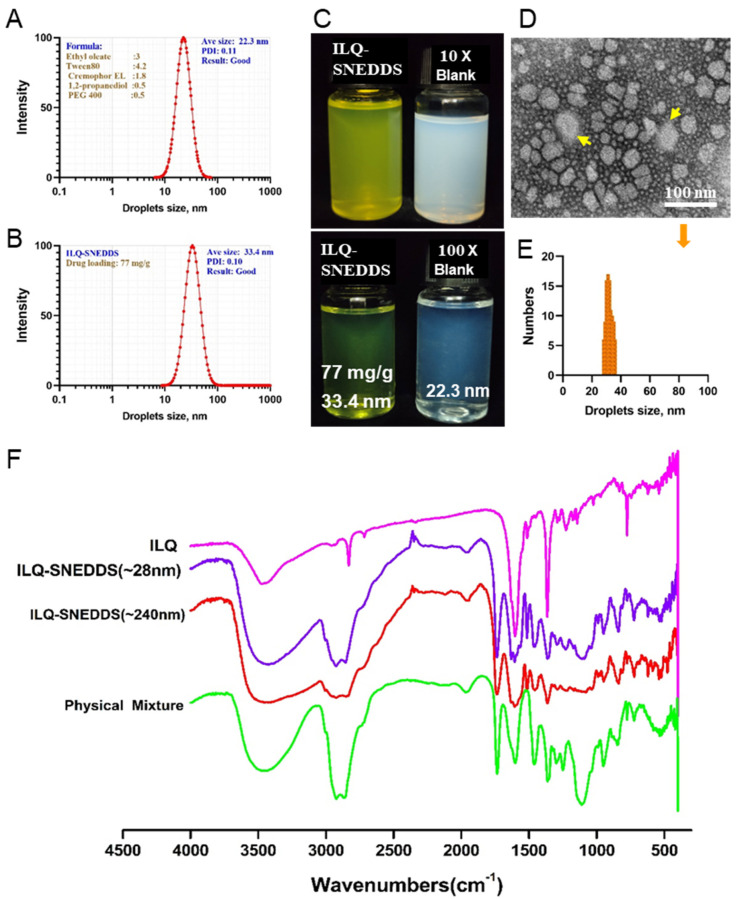
Physiochemistry characterization of ILQ-SNEDDS. (**A**) Blank SNEDDS droplet size distribution. (**B**) ILQ-SNEDDS droplet size distribution. (**C**) The appearance of freshly prepared Blank SNEDDS and ILQ-SNEDDS. (**D**) The TEM image of ILQ-SNEDDS (50 times dilution). The yellow arrows indicates that the droplets are at the edge of drug overload. (**E**) ILQ-SNEDDS droplet size distribution by TEM. (**F**) FTIR spectra of ILQ-SNEDDS (28 nm, 75 mg/g), ILQ-SNEDDS (240 nm, 135 mg/g), physical mixture (ILQ + SNEDDS), and ILQ.

**Figure 4 pharmaceuticals-15-01587-f004:**
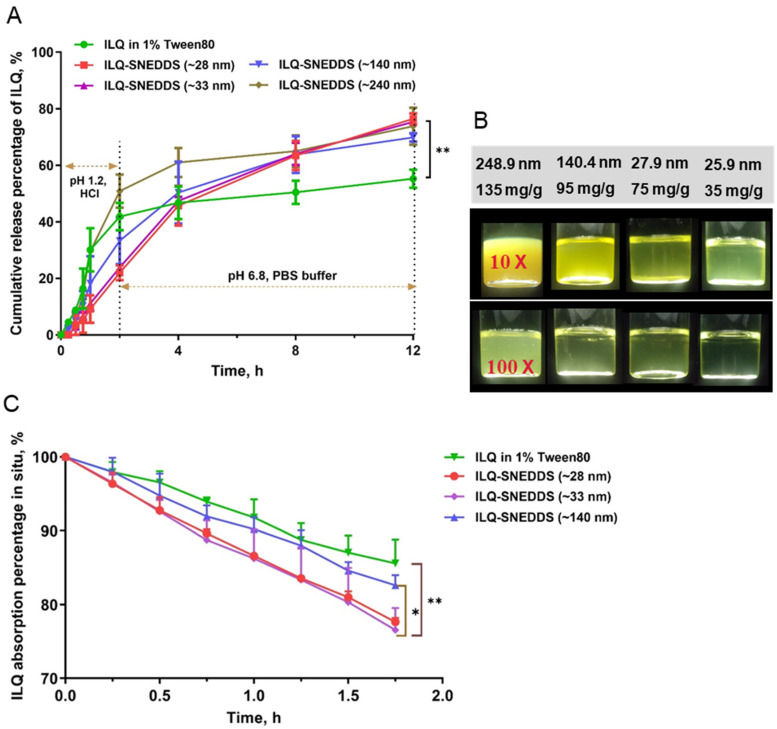
In vitro release profiles of ILQ from ILQ suspension and ILQ-SNEDDS with four different drug loading and their in-situ absorption. (**A**) In vitro release profiles of ILQ-SNEDDS in simulated gastric (pH 1.2, for the first 2 h) and intestinal (pH 6.8 PBS for the rest 22 h) fluid. (**B**) The appearance of freshly prepared ILQ-SNEDDS with different drug loading. (**C**) Absorption percentage curve of ILQ suspension or ILQ-SNEDDS with different drug loading in rats’ proximal jejunum segment after perfusing 2 h with an ILQ dose of 200 μg/mL. Length of jejunum: 12~15 cm; Each value represents the mean ± SD, * *p* < 0.05 and, ** *p* < 0.01 (Prism Mann–Whitney test, *n* = 3~4).

**Figure 5 pharmaceuticals-15-01587-f005:**
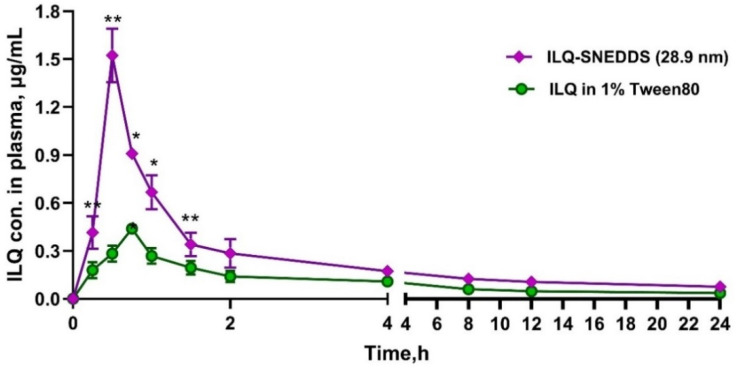
Mean plasma concentration–time profiles of ILQ in mice after oral administration of ILQ suspension or ILQ-SNEDDS (equivalent dose) at a dose of 35 mg/kg of ILQ. Each value represents the mean ± SD, * *p* < 0.05, ** *p* < 0.01. (Each time point, *n* = 4).

**Figure 6 pharmaceuticals-15-01587-f006:**
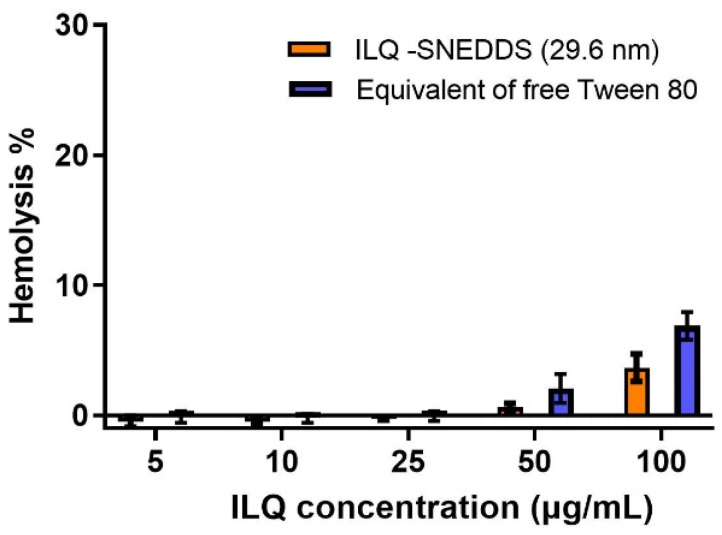
Hemolysis effect of ILQ-SNEDDS and equivalent amount of Tween-80 on rat RBC after incubation at 37 °C for 3 h (*n* = 3).

**Figure 7 pharmaceuticals-15-01587-f007:**
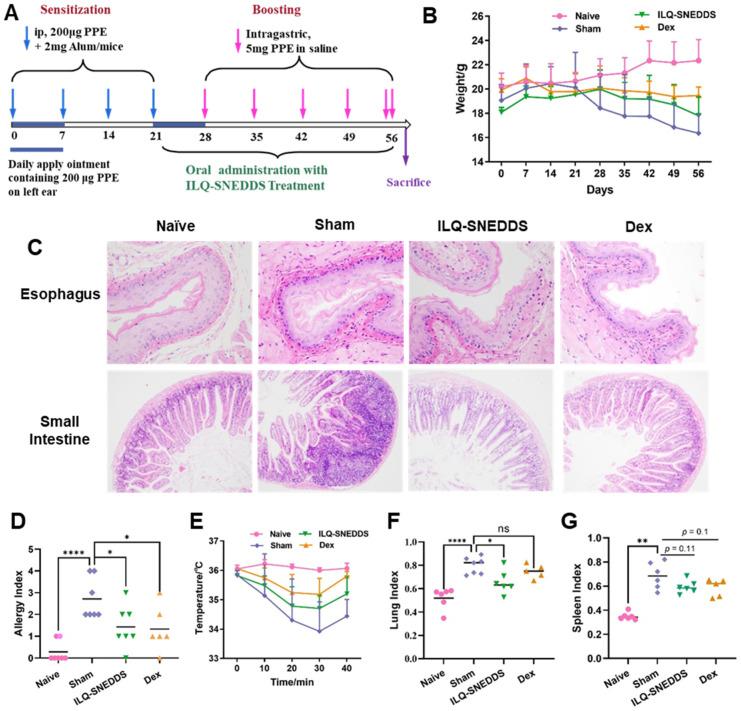
Daily oral treatment with ILQ-SNEDDS alleviated symptoms of food allergy-induced EoE-like disease mice. (**A**) Procedure for PPE-sensitized murine model of food allergy-induced EoE; (**B**) body weight; (**C**) H&E pathological section of esophagus (40×) and small intestine (20×) in each group. (**D**) Allergy index; (**E**) body temperature after the last challenge. (**F**) Spleen index, (**G**) lung index. Results are expressed as mean ± SD. *p* value was calculated by one-way ANOVA using Prism 9 software. * *p* < 0.05, ** *p* < 0.01, and **** *p* < 0.0001 compared to sham group.

**Figure 8 pharmaceuticals-15-01587-f008:**
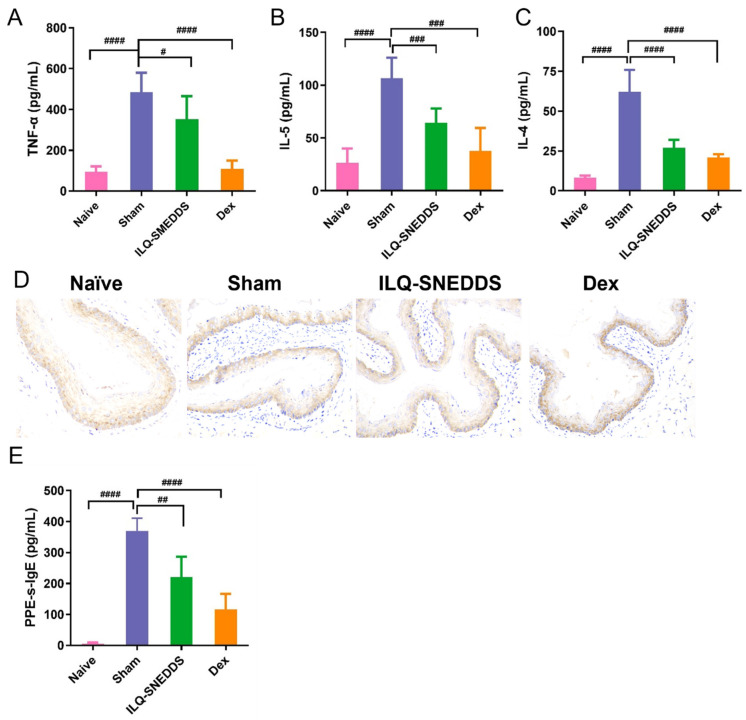
ILQ-SNEDDS treatment reduced the serum levels of the relevant Th2 inflammatory cytokines, TGF-β1 expression, and PPE-s-IgE. (**A**) TNF-α; (**B**) IL-4; (**C**) IL-5; (**D**) ICH for TGF-β1 expression (40×); (**E**) PPE-s-IgE. Results are expressed as mean ± SD. *p* value was calculated by one-way ANOVA using Prism 9 software., #, ##, ###, ####, *p* < 0.05, *p* < 0.01, *p* < 0.001, and *p* < 0.0001 as compared to the sham group.

**Table 1 pharmaceuticals-15-01587-t001:** Orthogonal experiments of three factors and three levels. Factor A: Percentage of oil phase; Factor B: Mixed Surfactant (Tween80: Cremophor EL) mass ratio; Factor C: the mass ratio of surfactant and co-surfactant.

Num	Factor A	Factor B	Factor C	Drug Loading mg/g	Ave Size (nm)	Ave PDI	ζ Potential (mV)
1	25%	9:1	6	52.59	19.14	0.160	−6.74
2	25%	8:2	2.5	76.06	237.3	0.231	−5.28
3	25%	7:3	3.7	31.92	16.89	0.152	−9.27
4	30%	9:1	2.5	51.41	11.99	0.159	−8.34
5	30%	8:2	3.7	44.4	15.21	0.115	−8.61
6	30%	7:3	6	73.06	28.39	0.178	−10.11
7	35%	9:1	3.7	46.3	29.63	0.200	−9.79
8	35%	8:2	6	37.53	13.25	0.158	−3.47
9	35%	7:3	2.5	54.88	190.06	0.204	−4.39
Levels	A	B	C				
1	53.52	50.10	56.03				
2	57.93	52.66	40.87				
3	46.24	54.93	60.78				
R	11.69	4.83	19.91				
Ranking	2	3	1				

**Table 2 pharmaceuticals-15-01587-t002:** The results of stability of ILQ-SMEDDS at different storage conditions.

Items		Droplet Size (nm)	PDI	ζ-Potential (mV)
Freshly prepared (0 day)		27.93	0.151	−9.84
Dilution with pH 6.8 PBS		28.85	0.191	/
Dilution with 0.01 M HCl		28.33	0.204	/
1 month	at 4 °C	28.89	0.206	−8.34
	at 37 °C	29.30	0.211	−8.35
2 months	at 4 °C	140.41	0.224	−8.12
	at 37 °C	241.48	0.237	−4.41

**Table 3 pharmaceuticals-15-01587-t003:** The absorption coefficient calculated in situ intestinal perfusion (*n* = 3 or 4).

Items	Globule Size and Drug Loading	Ap/%	Ka/h^−1^	Papp·10^−4^/h^−1^·cm^2^	Enhancement Ratio
ILQ-SNEDDS	28.9 nm (35 mg/g)	22.02 ± 0.39	0.144 ± 0.0043	29.58 ± 2.03	1.61
	33.6 nm (75 mg/g)	23.45 ± 2.96	0.149 ± 0.0060	30.97 ± 4.49	1.69
	140.4 nm (95 mg/g)	17.39 ± 1.34	0.104 ± 0.0107	20.14 ± 2.46	1.10
ILQ suspension	-	14.43 ± 3.27	0.086 ± 0.0080	18.34 ± 2.86	1.00

**Table 4 pharmaceuticals-15-01587-t004:** Pharmacokinetic parameters after oral administration of free ILQ (35 mg/kg) or ILQ-SNEDDS (Equivalent Dose) in SD rats. (* *p* < 0.05.).

Items	Free ILQ Suspension	ILQ-SNEDDS
K_e_, h^−1^	0.4 ± 0.01	0.36 ± 0.01
t_1/2_, h	1.79 ± 4.12	2.12 ± 6.87
T_max_, h	0.75 ± 0.00	0.5 ± 0.00 *
C_max_, μg/mL	0.43 ± 0.02	1.52 ± 0.13 *
AUC_0–24_, μg/mL·h	1.86 ± 0.12	3.76 ± 0.38 *
Vz/F, L	320.32 ± 56.30	170.11 ± 13.11 *

## Data Availability

Data is contained within the article.

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
