# Peer review of "Development of an Oral Isoliquiritigenin Self-Nano-Emulsifying Drug Delivery System (ILQ-SNEDDS) for Effective Treatment of Eosinophilic Esophagitis Induced by Food Allergy"

_pharmaceuticals, 2022, doi:10.3390/ph15121587_

Round 1

Reviewer 1 Report

Comments to Authors

1.      The key words are a little wordy

2.      Could you write a short introduction about nanoemulsions as an advanced drug delivery system

3.      What is Isoliquiritigenin structure? Why authors chose it? What is its pharmacological properties?

4.      Authors should run an encapsulation efficiency test to determine the amount of Isoliquiritigenin entrapped in the formulation

5.      The analysis used to determine particle size, morphology, polydiversity, surface charge, stability should be mentioned clearly in subheadings. Also please add the device model, calibrations and how the samples were prepared

6.      Please add more references to the methodology

7.      These references will be of benefit

·         Javed, R., Ghonaim, R., Shathili, A., Khalifa, S. A., & El-Seedi, H. R. (2021). Phytonanotechnology: A greener approach for biomedical applications. In Biogenic Nanoparticles for Cancer Theranostics (pp. 43-86). Elsevier.

Author Response

Dear professor,

 Thank you for the opportunity to revise our manuscript. We also appreciate you for the constructive comments. We have all effort and address the reviewed comments and revised our manuscript.

  Best Regards.

Mingzhuo

Our point-by-point responses to the reviewer’s comments are as follows:

  1. Response: We appreciate the reviewer for this valuable comment. We deleted ALLERGY.
  2. Response: We appreciate the reviewer for the valuable comment.

Nanotechnology-based drug delivery systems have done well work in increasing the solubility and bioavailability of insoluble drugs in vivo, which has gradually become a consensus. Self-nano-emulsifying drug delivery system (SNEDDS) have been paid extensive attention for its excellent property for oral administration, such as spontaneous formation in the gastrointestinal tract, ease of manufacture and low cost.

It reads in lines “75-80”

  1. Response: We appreciate the reviewers for this excellent comment.

We added the structure of ILQ in Fig 1.

 In our previous study, we found that ILQ is one of the strongest anti-asthma compounds isolated from Glycyrrhiza and has a strong eotaxin-1 inhibitory effect, with a low IC50 of 1.95 μg/mL.

It reads in lines 69-75.

  1. Response: We appreciate the reviewers for the valuable comment.

  Drug encapsulation efficiency (EE) and drug loading (DL) were measured by the centrifugation method previously reported in the literature with some modification (AAPS PharmSciTech, 2019. 20(5): p. 218). The newly prepared ILQ-SNEDDS) was centrifuged at 8000 rpm for 30 min (Microfuge 20R, Beckman Coulter, USA) to remove the unincorporated ILQ. The filtrate was diluted with methanol. The amount of ILQ was determined by HPLC after filtered through 0.22 μm filter membrane. The encapsulation efficiency was calculated as follows:

                               (2-1).

                                (2-2)

In which Wloaded, and Wtotal-added are the weight of encapsulated ILQ, the total amount of ILQ added, and the weight of emulsion added, respectively.

It reads in lines “143-151”.

The encapsulation efficiency was 92.50% ± 0.45. It reads in line “342”.

  1. Response: We appreciate the reviewers for the valuable comment.

We revised the relevant analysis mentioned.

  When the ILQ drug loading was lower than 77.9 mg/g, ILQ-SNEDDS could self-assemble into subspherical uniform droplets, with an average size of 30.5 ± 3 nm, a PDI of 0.10 ± 0.03, and a Zeta potential of -10.05 ± 1.0 mV. In situ intestinal absorption showed that SNEDDS (33 nm) significantly increased the apparent permeability coefficient of ILQ by 1.69 times, and the pharmacokinetic parameters also confirmed that SNEDDS sharply increased the plasma concentration and bioavailability of ILQ by 3.47 and 2.02 times, respectively.

It reads in lines “28-37” and “300-302”.

  1. Response: We appreciate the reviewers for the valuable comment.

   We added 15 more references in the manuscript.

  1. Response: We appreciate the reviewers for the valuable comment.

   It is a pity. Our school dose no such subscription, so we can't download it and read them. It is very kind of you to send this part of content to me if it is convenient for you. However, we searched another relevant literature for reading and learning, which really benefited us a lot. In view of the good anti-inflammatory and antioxidant effects of ILQ, it has significant potential in the treatment of asthma, food allergy and EoE. We will continue to look for other nanotechnology that can improve the bioavailability of ILQ more efficiently in our subsequent research.

It reads in lines “581-587”.

Reviewer 2 Report

This is well organized and written paper on development of optimized drug formation and evaluation of its properties. 

I do have only a few comments on the manuscript: 

1. the in vivo PK study has been conducted in feemale rats, which is  unusual. There should be an explanation, why male subjects have not been chosen and also a discussion section on how the sex dependent hormonal changes could affect the obtained results. 

2. the observed absorption differences between the two formulations should be discussed in the view of recent reports on high interindividual variability in animals (10.3390/pharmaceutics14030643) that could in reality mimick formulation performance. 

3. methodological limitation of the PD study is that there is no IQL free suspension group, so there is no evidence that the improved formulation brings any potential PD benefit. This needs to be clearly fomulated in the manuscript. 

Author Response

Dear professor,

 Thank you for the opportunity to revise our manuscript. We also appreciate you for the constructive comments. We have all effort and address the reviewed comments and revised our manuscript.

  Best Regards.

Mingzhuo

Our point-by-point responses to the reviewer’s comments are as follows:

  1. Response: We appreciate the reviewer for this excellent comment.

In Asia, EOE is much less prevalent than in western countries. In particular, EOE is still rare in China. In western countries, males are 3–4 times more likely to be diagnosed with EoE compared with females, despite having a similar gene expression, clinical presentations, histories of atopy, findings from endoscopy, and histologic characteristics. In both adults and children, there were no significant differences between sexes in mid/proximal or distal esophageal eosinophil count. (Clin Gastroenterol Hepatol. 2016 Jan;14(1):23-30. doi: 10.1016/j.cgh.2015.08.034.) There is almost no significant difference in the proportion of male/female cases reported in China. (Journal of Gastroenterology and Hepatology. 2015; 30 (Suppl. 1): 71–77).

So far, except Smad3-deficient and Sharpin-deficient EOE mice model, EOE animal models was mainly based on food-allergy or asthma induced murine model. In our study, the peanut protein extract (PPE)-induced EoE model was established using a previously reported method with minor modification. Female mice were used in this manuscript. (Sci Rep. 2018 Jan 18;8(1):1049.)

Eosinophilic esophagitis (EoE) is commonly associated with concomitant atopic diseases including atopic dermatitis (AD), food allergy and allergic airway (AA) diseases including asthma. There are many manuscripts reported female mice experience more airway remodeling and TH2 inflammation compared with male mice. (Pharmacol Res. 2019 Jan;139:182-190.; Clin. Exp. Allergy. 35 (2005) 1496–1503.; Respirology. 18 (2013) 797–806)

In order to make the study more meaningful, we would more reasonably consider the effect of mice gender in our follow-up study. We thank reviewers for this comment.

In reads in lines “588-596”

  1. Response: We appreciate the reviewer for this excellent comment.

However, due to the differences in hormone levels and metabolism between men and women, it should be well considered with regards to mice gender on EOE pathogenesis and drug absorption (Pharmaceutics, 2022. 14(3).).

In reads in line “596”

  1. Response: We appreciate the reviewer for this excellent comment.

Yes, no ILQ free suspension group is a disadvantage in the manuscript. In our previous study, we developed an ILQ-SNEDDS for asthma treatment, and the ILQ-SMEDDS group at the dose of 10 mg/kg showed better anti-asthma effect than that of the ILQ suspension group at the dose of 20 mg/kg. In addition, we found that ILQ is one of the most potent flavonoids isolated from Glycyrrhiza to significantly suppress Th2 type immune response and the production of eotaxin-1[14, 16, 17], both of which greatly contributes to eosinophil inflammation in EoE. Because of asthma, food allergy and EOE share the similar pathogenesis in clinic. Therefore, we made first attempt to use ILQ to treat EOE, choosing ILQ-SNEDDS with higher bioavailability instead of Free ILQ, which was also a limitation in the current study.

In reads in lines “480-484” and “599”.

Round 2

Reviewer 2 Report

Thank you for providing responses to my comments. I have no further issues.